# Investigation of MO Adsorption Kinetics and Photocatalytic Degradation Utilizing Hollow Fibers of Cu-CuO/TiO_2_ Nanocomposite

**DOI:** 10.3390/ma17184663

**Published:** 2024-09-23

**Authors:** George V. Theodorakopoulos, Sergios K. Papageorgiou, Fotios K. Katsaros, George Em. Romanos, Margarita Beazi-Katsioti

**Affiliations:** 1Institute of Nanoscience and Nanotechnology, National Center for Scientific Research “Demokritos”, 15341 Agia Paraskevi, Athens, Greece; s.papageorgiou@inn.demokritos.gr (S.K.P.); f.katsaros@inn.demokritos.gr (F.K.K.); 2School of Chemical Engineering, National Technical University of Athens, Zografou Campus, 9 Iroon Polytechniou Street, Zografou, 15772 Athens, Greece; katsioti@central.ntua.gr

**Keywords:** photocatalytic nanocomposites, hollow fibers, copper and copper oxide nanoparticles, titania, batch reactor, MO degradation, adsorption, kinetics study, regeneration, reusability

## Abstract

This comprehensive study explores the kinetics of adsorption and its photocatalytic degradation of methyl orange (MO) using an advanced copper-decorated photocatalyst in the form of hollow fibers (HFs). Designed to boost both adsorption capacity and photocatalytic activity, the photocatalyst was tested in batch experiments to efficiently remove MO from aqueous solutions. Various isotherm models, including Langmuir, Freundlich, Sips, Temkin, and Dubinin–Radushkevich, along with kinetic models like pseudo-first and pseudo-second order, Elovich, Bangham, and Weber–Morris, were utilized to assess adsorption capacity and kinetics at varying initial concentrations. The results indicated a favorable MO physisorption on the nanocomposite photocatalyst under specific conditions. Further analysis of photocatalytic degradation under UV exposure revealed that the material maintained high degradation efficiency and stability across different MO concentrations. Through the facilitation of reactive oxygen species generation, oxygen played a crucial role in enhancing photocatalytic performance, while the degradation process following the Langmuir–Hinshelwood model. The study also confirmed the robustness and sustained activity of the nanocomposite photocatalyst, which could be regenerated and reused over five successive cycles, maintaining 92% of their initial performance at concentrations up to 15 mg/L. Overall, this effective nanocomposite photocatalyst structured in the form of HF shows great promise for effectively removing organic pollutants through combined adsorption and photocatalysis, offering valuable potential in wastewater treatment and environmental remediation.

## 1. Introduction

Dyes have found widespread application in numerous industrial processes, including the production and use of textiles, food, papers, plastics, paints, cosmetics, and others [1]. Of all the industries, the textile sector is particularly notable for its significant use of water, dyes, and a range of organic and inorganic chemicals utilized throughout the textile production process. Methyl orange (MO), a frequently utilized acidic anionic azo dye [2,3], is extensively and consistently employed in textiles, the food industry, and laboratory research. However, it represents a potential hazard to aquatic ecosystems due to its toxicity to aquatic organisms [4] and could pose a risk to living systems, if present in high concentrations [5]. Thus, it is crucial to remove this dye from industrial effluent before it is released into the environment. To achieve this goal, a range of methods encompassing adsorption [6,7,8,9,10,11], membrane filtration [12,13,14,15,16], electrochemical techniques [17,18,19,20,21,22], solvent extraction [23,24], and photocatalytic degradation [25,26,27,28] have been utilized for wastewater purification and decontamination.

Among the aforementioned techniques, heterogenous photocatalysis is considered as a highly promising treatment and an exceptionally efficient approach for water purification. This assessment is based on the remarkable improvements in the degradation activity and efficiency of lately studied novel photocatalytic nanocatalysts, which leads to the complete mineralization and removal of contaminants frequently under visible light [29,30,31,32,33,34,35]. Furthermore, its cost-effectiveness is noteworthy, particularly regarding photocatalytic copper-titania nanocomposite materials due to the affordability, earth-abundance, and non-toxic characteristics of both components [36,37,38,39,40].

However, the utilization of these novel and efficient photocatalysts in powder form has been associated with a number of challenges. When utilizing the photocatalyst in powder form within a slurry system, it becomes necessary to separate, recycle, and recover it from the treated wastewater, a process that can be time-consuming and costly [41,42]. Photocatalyst powders also tend to agglomerate, resulting in a reduced surface area and decreased photocatalytic reactivity [43]. Another issue is that the efficiency of light utilization decreases because the penetration depth of UV or visible light is restricted by significant absorption by both the catalyst particles and the dissolved pollutants [41,44]. Some types of photocatalytic semiconductors such as CdS, ZnO, and BiVO_4_ are susceptible to degradation over time, primarily due to processes such as photocorrosion or surface oxidation [45,46,47], which can diminish their effectiveness. Moreover, the mass transfer of reactants to the catalyst surface is generally slower when these photocatalysts are used as powders [45,48,49]. This reduced mass transfer can hinder the overall efficiency of the photocatalytic process by limiting the availability of reactants at the photocatalyst’s active sites, especially when these are active sites of composite nanocatalysts. In this context, for practical applications, the photocatalytic nanocomposite materials are usually structured in macroscopic scaffolds (beads [50,51,52,53,54,55,56], pellets, and pearls) by embedding and extruding them within polymer melts. A novel approach for shaping photocatalysts involves extruding them into porous hollow fibers, which, due to their unique morphological and structural characteristics, provides better mass transfer, light utilization, etc. [57,58,59,60,61,62].

In our recent study [62], we elaborated a series of novel developed photocatalysts shaped in the form of HFs, in relation to their capacity to adsorb and photocatalytically eliminate MO dye, a representative anionic contaminant. After conducting fast-screening tests to rank the adsorption and photocatalytic capacities of the developed materials in the form of HFs, we identified Cu-decorated nanocomposite as the most promising candidate. This material is now subjected to a more detailed and extensive study, focusing on varying initial MO concentrations and dissolved oxygen conditions. Considering that one of the supplementary goals of this research is to elucidate the role of oxygen in both capturing photoinduced electrons and producing active oxygen radicals, experiments were conducted employing O_2_-saturated and O_2_-depleted solutions. This approach facilitated the determination of whether the favorable or unfavorable outcomes associated with dissolved oxygen in photocatalysis could be exclusively attributed to its role as an electron scavenger, or if it also plays a role in pollutant adsorption, potentially contributing to a significant synergistic enhancement of photocatalytic performance.

Furthermore, this study explored adsorption parameters using various isotherm models (Langmuir, Freundlich, Sips, Temkin, and Dubinin–Radushkevich) along with kinetic models like pseudo-first and pseudo-second order, Elovich, Bangham, and Weber–Morris. This investigation aimed to shed light on the adsorption mechanism pathway on photocatalytic nanocomposites, evaluate the photocatalytic capacity and efficiently formulate the photocatalytic system by selecting the most suitable model that aligns with the experimental kinetic data. Finally, the photocatalytic degradation process was thoroughly analyzed using the Langmuir–Hinshelwood model to understand the reaction kinetics and mechanisms. Alongside this, the study also examined the robustness and long-term activity of the copper-titania nanocomposite, focusing on its ability to maintain consistent performance over prolonged use and assessing their potential for practical applications in wastewater treatment and environmental remediation.

## 2. Materials and Methods

All reagents used were of analytical quality and were used straight without additional purification. Sodium alginate [SA, (C_6_H_7_NaO_6_)_n_] was obtained from Sigma-Aldrich Chemie GmbH (Buchs, Switzerland) and TiO_2_ (P25) from Degussa AG (Essen, Germany). Glutaraldehyde (C_5_H_8_O_2_, Grade II) was sourced from Acros Organics (Geel, Belgium), NaOH (99.8%) and Cu(NO_3_)_2_·3H_2_O (99%) from Merck KGaA (Darmstadt, Germany), citric acid (99.5%) from Riedel-de Häen (Seelze, Germany), and ethanol (CH_3_CH_2_OH, 99.8%) from Fisher Scientific UK Ltd. (Loughborough, UK). Methyl Orange (MO, C_14_H_14_N_3_NaO_3_S, 99%, Sigma-Aldrich, St. Louis, MO, USA) dye was employed as the water pollutant for the photocatalytic study of the prepared hollow fibers.

### 2.1. Hollow Fibers Preparation

The precursor hollow fibers (HFs) of the Cu-CuO/TiO_2_ nanocomposite were prepared by the wet spinning, cross-linking process in a spinning set-up, as described in detail in our previous study [62]. In brief, the HFs were spun by extruding the alginate/TiO_2_ slurry through a tube in orifice spinneret into an ethanol, glutaraldehyde, and HCl (5 N) mixture (90.16%, 8%, and 1.84% vol, respectively), which also served as the bore liquid. After the spinning process, the HFs were stored in the same solution. To further introduce copper ions in the system, the HFs were immersed in copper nitrate solution (1% in ethanol) for 24 h, dried overnight, and then pyrolyzed under N_2_ flow at 600 °C for 6 h. Cu^2+^ ions were incorporated to achieve the formation of an all-ceramic HF made of the Cu-CuO/TiO_2_ nanocomposite. This process concluded to a Cu-CuO/TiO_2_ nanocomposite having well dispersed copper/copper oxide nanoparticles, because of the effective adsorption copper ions by alginate carboxy groups. Moreover, this method does not involve direct anchoring of copper ions with TiO_2_ surface before pyrolysis. Instead, copper ions are chelated with the carboxyl groups of sodium alginate. Therefore, it can be stated that pyrolysis resulted to a nanocomposite photocatalyst rather than to a heteroatom (Cu)-doped photocatalyst.

### 2.2. Characterization Techniques

In brief, the pore structural and textural, along with the morphological and chemical, properties of the Cu-CuO/TiO_2_ nanocomposite were investigated using liquid nitrogen (LN_2_) adsorption–desorption isotherms at 77 K on an automated volumetric system (AUTOSORB-1, Quantachrome Instruments, Boynton Beach, FL, USA), SEM (Jeol JSM-7401F Field Emission Scanning Electron Microscope equipped with Gentle Beam mode, Tokyo, Japan) and EDS (Xplore-15 SDD detector, Oxford Instruments, High Wycombe, UK) analyses. XRD diffraction patterns were recorded using a Rigaku R-AXIS IV Imaging Plate Detector mounted on a Rigaku RU-H3R Rotating Anode X-ray Generator (Rigaku Corporation, Tokyo, Japan), and Raman spectra were obtained with a Renishaw inVia Reflex system in backscattering configuration, using an Ar^+^ ion laser (514.5 nm) as the excitation source, with the laser power adjusted to 0.5 mW/μm^2^ on a Leica DMLM microscope (Renishaw, Wotton-under-Edge, UK). The total procedure is detailed in our previous work [62].

### 2.3. Photocatalytic Batch Procedure of the Developed Nanocomposite Material

To evaluate the photocatalytic efficacy of the developed nanocomposite material, a batch process was employed under controlled experimental conditions. Before each experiment, aqueous methyl orange (MO) solutions (ultrapure water, Milli-Q, 18 MΩ·cm) were sparged with either oxygen or inert gas for 3 h, resulting in oxygen-saturated and oxygen-depleted conditions, respectively, allowing the evaluation of dissolved oxygen’s impact on photocatalytic activity. Detailed information on the photocatalytic experimental procedure and the total assessment can be found in our previous work [62]. In short, the photocatalytic experiments were conducted in a borosilicate glass cell encompassing 30 mL of MO solution and 75 mg of fibers (with an outer diameter of 700 μm and a wall thickness of 150 μm) cut into 5 mm segments placed 5 cm from four UV-A lamps (350–390 nm) in a custom-made black box photoreactor with uniform illumination ensured. Before being exposed to UV radiation, the solutions were magnetically agitated in the absence of light for 105 min to establish adsorption equilibrium, with the initial concentration measured to account for dark-phase adsorption. A small aliquot of the MO solution was withdrawn at specific time intervals for measurement and then returned to the solution. Initial MO concentrations of 6.3, 10, 12, 15, 18, and 24 mg/L were used to elucidate the effect of varying MO concentrations. MO concentrations were determined using a Hitachi U-3010 UV-visible spectrophotometer (Tokyo, Japan). Calculations were based on the characteristic absorbance peaks (464 nm for azo group and 271 nm for aromatic ring), following Beer–Lambert law. Finally, in order to assess the photocatalytic efficiency and durability of the developed HFs while exposed to UV radiation, a sequence of five consecutive experiments was conducted. Hence, after the end of each experimental cycle, the used sample was retrieved from the spent MO solution, subjected to rinsing twice with ultrapure water, dried, and then used again with a fresh quantity of MO solution from the stock solution for the next experimental cycle.

The calculation of the equilibrium adsorption of MO onto the photocatalyst (q_e_, mg/g) was determined from the subsequent equation:(1)$$q_{e}=\left(C_{0}-C_{e}\right)\times \frac{V}{w}$$
where C_0_ is the initial MO concentration (mg/L), C_e_ is the equilibrium MO concentration (mg/L), V is the volume of the solution (L), and w is the weight of the HF as the adsorbent (g).

The equation used to obtain the photodegradation efficiency R (%) is as follows:(2)$$R=\frac{\Delta C}{C_{e}}\left(\%\right)=\left(\frac{C_{e}-C_{t}}{C_{e}}\right)\times 100\%$$
where C_t_ is the MO concentration at any time during the experiment (mg/L).

## 3. Results

### 3.1. Characterization Results

As aforementioned in Section 2.2, the pore structure, texture, morphology, and chemical properties of the Cu-CuO/TiO_2_ nanocomposite were thoroughly analyzed using a variety of advanced characterization techniques. Scanning electron microscopy (SEM) was employed to investigate the surface and texture morphology of the HFs (Appendix A), while energy-dispersive X-ray spectroscopy (EDS) provided insight into the elemental composition and distribution across the nanocomposite (Appendix A). Additionally, X-ray diffraction (XRD) was used to examine the crystallographic structure (Appendix A), and Raman spectroscopy offered detailed information on the molecular and chemical bonding characteristics (Appendix A). The comprehensive analysis, including the resulting data from these techniques, is provided in the Appendix A.

### 3.2. Evaluation of MO Adsorption Capacity

To assess the MO adsorption capacity of the Cu-CuO/TiO_2_ nanocomposite, a range of initial MO concentrations was firstly tested in O_2_-saturated solutions, as illustrated in Figure 1a. The MO adsorption capacity ranged from 2.3 mg/g to 6.7 mg/g at the highest concentration (24 mg/L). As previously illustrated in our prior work [62], decorating HFs with nanoparticles of zero-valent copper and/or copper oxide improved their pore structural and surface texture characteristics leading to an enhanced MO adsorption capacity. In addition, the presence of residual carbon enhances the adsorption capacity through π-π dispersion interactions between the MO and the photocatalyst [63,64], as well as the electron donor–acceptor complex mechanism [65,66,67]. This condition serves a dual purpose: first, it results in the accumulation of pollutant on the photocatalyst’s surface, and second, it weakens the bonds of the adsorbed molecules [64].

As shown in Figure 1a, MO adsorption increases over time for all concentrations, with the process being rapid initially and then slowing down as it approaches equilibrium. Higher initial MO concentrations result in greater adsorption capacities. Specifically, the adsorption curves for higher concentrations (18 and 24 mg/L) rise more steeply and achieve higher adsorption values compared to lower concentrations (6.3 and 10 mg/L). Higher concentrations also take longer to reach equilibrium compared to lower concentrations. It may be inferred that the adsorption process is concentration-dependent, with higher concentrations leading to higher adsorption capacities and a slower approach to equilibrium.

As presented in Figure 1b, for the O_2_-depleted MO solution (6.3 mg/L), the Cu-CuO/TiO_2_ nanocomposite exhibited a similar adsorption capacity of 2.28 mg/g, which increased to 5.7 mg/g at the highest concentration of 18 mg/L. Similarly, the MO adsorption increases over time for all concentrations, with the adsorption rate being rapid at first and then slowing as equilibrium is reached, consistent with the behavior observed under O_2_-saturated conditions. Higher initial MO concentrations lead to greater adsorption capacities, with the curves for higher concentrations (15 and 18 mg/L) rising more steeply and reaching higher adsorption values than those for lower concentrations (6.3 and 10 mg/L). Equilibrium is approached within 100 min, though the rate of adsorption slows over time. Higher concentrations tend to take longer to reach equilibrium compared to lower concentrations. On the other hand, the saturation points are reached more quickly in O_2_-depleted solutions, which generally exhibit a similar or even higher adsorption capacity compared to O_2_-saturated solutions.

The faster saturation and higher or similar adsorption capacity in O_2_-depleted solutions may be ascribed to the reduced competition between oxygen molecules and MO molecules for adsorption sites [68] on the Cu-CuO/TiO_2_ nanocomposite. In the absence of oxygen, more active sites are available for MO adsorption, leading to a quicker achievement of equilibrium. Additionally, the absence of oxygen might reduce the potential for oxidative degradation of MO, allowing more of the dye to be adsorbed, thereby increasing the overall adsorption capacity. In the context of O_2_-depleted solutions, the absence of oxidative reactions helps maintain the integrity and reactivity of the functional groups on the HFs, as discussed in our previous work [62], which contributes to higher adsorption capacities. Consequently, the quicker saturation in O_2_-depleted solutions results from the more efficient utilization of adsorption sites and the preservation of the adsorbent’s surface properties, leading to an enhanced adsorption performance.

The adsorption isotherm for a solid–liquid system is an important physicochemical characteristic that reveals the adsorption characteristics and performance and provides insights into the catalyst’s surface properties. In this study, five well-established isothermal models—Langmuir, Freundlich, Sips, Temkin, and Dubinin–Radushkevich (D–R)—were used to analyze the experimental data for both O_2_-saturated and O_2_-depleted solutions. The Langmuir model assumes monolayer adsorption on the adsorbent’s surface without attraction or interaction between the adsorbed molecules [69]. As per this model, the adsorption isotherm is expressed by the Equation (S1). In addition, the parameter maximum surface coverage (θ_max_) is frequently used in adsorption studies to describe the maximum fraction of a surface covered by adsorbate molecules at equilibrium and it is expressed by the following equation [70]:(3)$$\theta_{max}=\frac{b\times C_{max}}{1+b\times C_{max}}$$This equation demonstrates how surface coverage depends on the balance between the adsorption rate, characterized by b, and the concentration of the adsorbate. Moreover, the essential characteristics of Langmuir adsorption isotherm can be expressed concerning a dimensionless constant known as the separation factor or equilibrium parameter (R_L_), which is determined by the Equation (S2) [71].

Subsequently, the adsorption behavior of MO on the Cu-CuO/TiO_2_ nanocomposite was studied by the Freundlich isotherm model. The Freundlich isotherm, being an empirical equation, postulates that the adsorption surface exhibits heterogeneity, and that the adsorption process extends beyond the conventional formation of a monolayer. The Freundlich isotherm [72] Equation (S3) is represented in the Appendix A. Another empirical adsorption equation commonly employed for the interpretation of adsorption data is the Sips model. The proposed model can be seen as a synthesis of the Langmuir and Freundlich equations [73] and it is expressed by the Equation (S4).

Temkin isotherm model considered that the effect of indirect adsorbate interactions on the adsorption process and suggested that because of these interactions, the heat of adsorption among all the molecules in the layer would exhibit a linear reduction with the increase in surface coverage. Additionally, the model assumed that bonding energies are uniformly distributed up to certain binding energies [74]. The Temkin isotherm is expressed in the Formulation (S5). Finally, the D–R isotherm model with a high degree of regularity is an empirical adsorption model commonly used to describe the adsorption process on heterogeneous surfaces with Gaussian energy distribution [75]. The model is based on a semi empirical equation by which adsorption occurs through a mechanism of pore filling. It is a fundamental equation that qualitatively characterizes the adsorption of gases and vapors on microporous sorbents, assuming a multilayer character incorporating van der Waals forces, applicable for physical adsorption approaches. The isotherm is defined by the Equations (S6) and (S7).

At this stage, it is noteworthy that the preference for using non-linear models over linear ones in fitting adsorption processes arises from several key factors. Non-linear regression preserves the inherent mathematical relationships within the models, leading to more accurate parameter estimation. In contrast, linearization can distort the data by requiring transformations, potentially introducing errors and biases [76,77,78,79], and reducing the reliability of the model fit, especially in the context of complex adsorption processes. Non-linear fitting also enables the direct optimization of model parameters, providing a more precise and realistic representation of the adsorption behavior. This approach is crucial for effectively capturing the thermodynamics, kinetics, and mechanistic details of adsorption processes, thereby ensuring the precise design and optimization of adsorption systems [80,81].

Appendix A displays the experimental isotherms of MO adsorption along with the non-linear fitting of the five isotherm models in both O_2_-saturated and O_2_-depleted solutions. In each case, the isotherm models exhibit varying degrees of fitting to the experimental data, with some models aligning more closely than others. In the O_2_-saturated solutions (Appendix A), the Langmuir and Sips models appear to provide the best fit to the experimental data suggesting that the adsorption process might involve characteristics of both homogeneous and heterogeneous adsorption sites. The Freundlich model also fits well, but deviates slightly, especially at higher equilibrium concentrations C_e_, indicating some heterogeneity. Similarly, the D–R model shows reasonable agreement but deviate slightly at higher concentrations. However, the Temkin model deviates more significantly at lower concentrations. Conversely, in the O_2_-depleted solution (Appendix A), the Langmuir and Sips models once again illustrate the most accurate correspondence to the experimental data, with the Freundlich model following closely behind. The Temkin and D–R models exhibit greater deviations across the concentration range. Certainly, in this case, it would be helpful to have more experimental data to gain a clearer and better understanding of how all the models fit. Overall, the Langmuir and Sips models consistently provide the best fit under both conditions, suggesting they may be the most appropriate models for describing MO adsorption under varying oxygen conditions.

The fitting parameters for all models have been derived from the isotherms, with their values presented in Appendix A. The symbol o indicates the parameters for the O_2_-saturated MO solutions, while i denotes those for the O_2_-depleted MO solutions. The adsorption isotherm parameters for MO uptake onto the Cu-CuO/TiO_2_ nanocomposite indicate that the Langmuir and Sips models provide the best fit for both O_2_-saturated and O_2_-depleted solutions, with high coefficient of determination R^2^ values of 0.999 and 0.906, respectively. These models suggest that the adsorption process follows monolayer coverage on a homogeneous surface, with the HFs having binding sites thar are relatively uniform nature, since a value of 1 for n reduces the Sips model to the Langmuir model. The Langmuir model’s maximum adsorption capacities (q_m_) are 20.73 and 22.80 mg/L, respectively, indicating that the adsorption efficiency is slightly higher in O_2_-depleted conditions. In contrast, the Freundlich and Temkin models, which in general are employed for adsorption on heterogeneous surface, have lower R^2^ values, particularly in the O_2_-depleted solutions, indicating a less accurate fitting. The Freundlich model’s n values of 1.28 and 1.17 suggest relatively constant adsorption intensity across conditions, falling within the favorable adsorption range (1 < n < 10). These data indicate that with an increase in the solute (MO) concentration, the adsorbent surface becomes more effective at capturing MO molecules. This behavior is typical of physical adsorption, where adsorption sites may vary in affinity, allowing for multilayer adsorption on the adsorbent surface. Values approaching 1 suggest a rather homogeneous distribution of adsorption sites, reflecting moderately strong and consistent adsorption intensity across the adsorbent surface. The Temkin model, with K_To_ = 0.949 L/g and K_Ti_ = 1.091 L/g, suggests moderate variations in adsorption heat. Since these values are positive, with b_T_ = (−ΔH), the adsorption reaction is exothermic. The D–R model suggests a lower adsorption capacity with q_so_ = 6.96 mg/g and q_si_ = 6.86 mg/g, showing that pore-filling adsorption is less significant than monolayer adsorption in this system. The MO adsorption on a copper/copper oxide nanoparticles-decorated, structured photocatalytic TiO_2_ system, which also bears residual carbon, is characterized as favorable physisorption, as the obtained mean free energy E values do not exceed 8 kJ/mol, indicating a physisorption process dominated by weaker van der Waals forces. Despite the various phases exposed to the water matrix (copper/copper oxide nanoparticles, carbonaceous phase, and TiO_2_ nanoparticles), the models suggest homogeneous binding sites for MO. Overall, the results underscore the effectiveness of monolayer adsorption, with the Langmuir model being the most representative of the adsorption process. The outcomes of the model fitting suggest that favorable physisorption phenomena occur between the MO pollutant and the HF photocatalyst under the specified conditions.

Another important parameter derived from the Langmuir isotherm, which aids in comprehending the adsorption mechanism and molecular interactions on the adsorbent surface, is the surface area (σ) occupied by a single molecule of the adsorbate on the adsorbent surface. One can compute this using the subsequent equation [82]:(4)$$\sigma =\frac{S_{BET}}{q_{max}\times N_{A}}$$
where σ is the surface area occupied by a single molecule of adsorbate (Å^2^), S_BET_ is the specific surface area determined by the BET (Brunauer-Emmett-Teller) method (m^2^/g), q_max_ is the maximum monolayer adsorption capacity (mg/g), and N_A_ is the Avogadro’s number (6.022 × 10^23^ molecules/mol).

The occupied surface area σ was calculated to be 162.3 Å^2^ in O_2_-saturated solutions and 147.6 Å^2^ in O_2_-depleted ones (with an S_BET_ of 61.9 m^2^/g calculated by LN_2_ porosimetry). This difference suggests that the presence of oxygen influences the orientation or packing of MO molecules on the surface of the Cu-CuO/TiO_2_ nanocomposite. In O_2_-saturated environments, the larger occupied area per molecule may indicate a less compact arrangement, potentially due to repulsive interactions facilitated by oxygen or a specific orientation that enhances surface interactions with available oxygen molecules. Conversely, in O_2_-depleted conditions, the smaller occupied area implies a more tightly packed arrangement, possibly because the absence of such interactions allows the molecules to occupy less space and pack more closely together. These observations suggest that the presence of oxygen influences the adsorption behavior of MO, potentially enhancing its effectiveness as a dye under varying oxygen conditions.

Appendix A presents the R_L_ and θ_max_ values for MO uptake in both O_2_-saturated and O_2_-depleted solutions at different initial concentrations. As shown, the R_L_ values are between 0 and 1, indicating that the MO adsorption from aqueous solutions onto the sample is favorable under the conditions used in this study. Moreover, the data show a consistent decrease in R_L_ values as the initial concentration of MO increases, suggesting that adsorption becomes less favorable at higher concentrations. This trend is observed in both O_2_-saturated and O_2_-depleted conditions, with the O_2_-saturated conditions generally exhibiting slightly higher R_L_ values than the O_2_-depleted conditions. The θ_max_ parameter, representing maximum surface coverage, increases with concentration in both conditions, indicating that more surface area is occupied by the dye. In both cases, θ_max_ presents similar trends with some fluctuation at higher concentrations, regardless of whether the solutions are O_2_-saturated or depleted. This suggests that the presence of oxygen is not a definite factor for MO adsorption efficiency, as previously discussed.

Finally, a concise analysis of the pore structural properties of the Cu-CuO/TiO_2_ nanocomposite at the end of this section would be beneficial. Figure 2 shows the LN_2_ porosimetry isotherm for the material revealing a type IV isotherm with a clearly observed hysteresis loop, characteristic of mesoporous materials [83]. The adsorption volume significantly increases at higher relative pressures (P/P_0_), indicating the existence of prominent mesopores or macropores. The inset of Figure 2 displays the pore size distribution (PSD), which was obtained from the LN_2_ isotherm desorption branch using the non-local density functional theory (NLDFT) method. The NLDFT analysis employed an equilibrium kernel for silica as the adsorbent and N_2_ 77 K as the adsorbate. The PSD presents a range of pore widths predominantly between 10 and 70 nm, with notable peaks around 10–15 nm and 30–40 nm, and a broader peak near 45 nm, indicating a wide distribution of pore sizes within the mesopore range. The extension up to 70 nm indicates a significant presence of macropores on the HFs. This suggests that the copper decoration and residual carbon on the HFs have created a hierarchical porosity structure, enhancing the adsorption capacity of MO by effectively facilitating the surface adsorption and diffusion of MO molecules.

### 3.3. Study of MO Adsorption Kinetics

The kinetics of adsorption of the Cu-CuO/TiO_2_ nanocomposite were examined employing pseudo-first-order and pseudo-second-order kinetic equations, along with Elovich, Bangham, and Weber–Morris kinetic models. The pseudo-first order (PFO) kinetic model is often employed to describe the adsorption kinetics of solutes from a liquid phase onto a solid surface. The underlying assumption of this model is that the occupancy rate of adsorption sites is directly proportional to the quantity of vacant sites. The mathematical representation of the PFO kinetic model is described by the Lagergren equation [84]:(5)$$q_{t}=q_{e}\times \left(1-e^{-k_{1}\times t}\right)$$
where q_t_ and q_e_ are the amount of adsorbate adsorbed at time t and at equilibrium, respectively (mg/g), and k_1_ is the pseudo-first order rate constant (min^–1^).

Conversely, the pseudo-second order (PSO) kinetic model assumes that the rate of adsorption is directly related to the square of the number of unoccupied sites. This model is expressed by the following equation [85]:(6)$$q_{t}=\frac{k_{2}\times q_{e}^{2}\times t}{\left(1+k_{2}\times q_{e}\times t\right)}$$
where k_2_ is the pseudo-second order rate constant (g/mg·min).

The Elovich kinetic model is commonly used to describe the kinetics of chemisorption processes on heterogeneous surfaces [86,87,88,89]. It is particularly applicable when the adsorption process involves a heterogeneous surface with a variety of activation energies. The Elovich model postulates that the rate of adsorption exhibits exponential decline as the quantity of adsorbate first adsorbed increases. The Elovich equation is given by [90]:(7)$$\frac{dq}{dt}=\alpha \times e^{-\beta \times q_{t}}$$The integrated form of Equation (7) can be written as follows:(8)$$q_{t}=\frac{1}{\beta }\times \ln\left(\alpha \times \beta \right)+\frac{1}{\beta }\times \ln t$$
where α, β are constants.

Furthermore, the kinetic data can be utilized to determine if pore diffusion is the sole rate-controlling step in the adsorption system by applying the Bangham’s equation [91,92]:(9)$$\log\left[\log\left(\frac{C_{i}}{C_{i}-m{\;\times \;q}_{t}}\right)\right]=\log\left(\frac{k_{0}\times m}{2.303\times V}\right)+a\times \log t$$
where C_i_ is the initial MO concentration (mg/L), m is the weight of the photocatalyst as the adsorbent (g), q_t_ is the MO amount retained at any time t (mg/g), V is the volume of the MO solution (mL), and k_0_ and a (<1) are constants.

Finally, to examine the diffusion mechanism and identify the rate-controlling steps influencing the adsorption kinetics [93], the kinetic data were analyzed using the Weber–Morris intraparticle diffusion model as described by the following equation [94]:(10)$$q_{t}=k_{id}\times t^{0.5}+C$$
where k_id_ is the intraparticle diffusion rate constant (mg/g·min^1/2^) and the intercept C is related to the thickness of the film boundary layer (mg/g). A plot of q_t_ versus t^0.5^ should produce a straight line with a slope of k_id_ and an intercept of C.

Figure 3 displays the experimental data utilized to fit the non-linear PFO and PSO kinetic models for the adsorption of MO in solutions that are saturated with O_2_ and depleted of O_2_. The kinetic parameters values obtained from these models are listed in Table 1.

Likewise, the experimental data collected at various MO concentrations were fitted to the Elovich, Bangham, and Weber–Morris kinetic models (Figure 4). The calculated values of the kinetic parameters are shown in Table 1. In Figure 4, the linear relationships depicted by the Elovich model are evident, with each line’s slope and intercept differing based on the concentration and oxygen conditions in the solutions. In both cases, higher concentrations generally result in steeper slopes, indicating faster adsorption rates.

The consistent linearity of the Bangham plots for all investigated MO concentrations indicates that the adsorption kinetics of MO are constrained by the pore diffusion of MO uptake onto the Cu-CuO/TiO_2_ nanocomposite [95,96]. The model effectively captures the diffusion dynamics within the pores of HFs. Variations in the slope and intercept values suggest that, although the mechanism remains consistent, the adsorption efficiency and capacity change with concentration. The adsorption characteristics exhibit variation with the initial concentration, thereby suggesting that the adsorption capacity and rate are subject to interference by the concentration of MO. This implies that at higher concentrations, there might be a higher initial rate of adsorption due to a greater driving force, but the overall process remains diffusion-controlled. Additionally, as observed in Figure 4 (third column), the linear graphs obtained for all concentrations investigated intersect the origin of the axis, indicating that intraparticle diffusion is the single rate-limiting step [92,97] and governs the MO uptake process. These linear regions of Weber–Morris plots correspond to mesopore and macropore diffusion onto the Cu-CuO/TiO_2_ nanocomposite, which represent the readily accessible adsorption sites on the adsorbent surface [91].

From the kinetic analysis of MO adsorption on the Cu-CuO/TiO_2_ nanocomposite under both O_2_-saturated and O_2_-depleted conditions, as summarized in the Table 1, the PSO model generally provides a higher correlation of determination R^2^, indicating a closer fit to the experimental data. However, the significant discrepancy between the predicted and experimental q_e_ values is a critical issue. The PSO model predicts much larger q_e_ values than those observed experimentally, suggesting that the model may overestimate the adsorption capacity in this system. This discrepancy could imply that the model’s assumption of chemisorption as the primary mechanism may not fully reflect the actual dynamics of MO adsorption on the Cu-CuO/TiO_2_ nanocomposite. In contrast, the PFO model, despite having slightly lower R^2^ values compared to the PSO model, provides q_e_ values that are much closer to the experimental data across various MO concentrations. This indicates that the PFO model, which postulates that the rate of adsorption sites occupancy varies in direct proportion to the quantity of vacant sites (typically associated with physisorption), might better represent the actual adsorption process for MO on the HFs. Given this, the PFO model may be preferable for describing MO adsorption on the Cu-CuO/TiO_2_ nanocomposite, as it offers a more accurate prediction of adsorption capacity, even though its coefficient of determination is slightly lower. The closer alignment of calculated and experimental q_e_ values implies that the adsorption process may be better described by a physical adsorption mechanism or a mixed mechanism, rather than the purely chemisorption suggested by the PSO model.

Additionally, the rate constant k_1_ generally remained stable, with slight fluctuations, across the initial MO concentrations in O_2_-saturated solutions. However, in O_2_-depleted solutions, the rate constant gradually increased, as the MO concentration increased. The observed stability of the rate constant k_1_ in O_2_-saturated solutions suggests that the presence of oxygen provides a consistent environment for the adsorption process, resulting in a same rate across different MO concentrations. Oxygen, by acting as an electron acceptor, may stabilize the photocatalyst’s surface and facilitate the interaction between the MO molecules and the active sites on the HFs surface. This stability indicates that oxygen helps maintain a balance in the adsorption kinetics, preventing significant variations in the rate constant despite changes in MO concentration. The minor fluctuations observed could be due to slight variations in experimental conditions or inherent surface heterogeneities, which do not significantly affect the overall adsorption kinetics when oxygen is present.

Conversely, the gradual increase in the rate constant k_1_ with rising MO concentration in O_2_-depleted solutions indicates that the absence of oxygen modifies the dynamics of the adsorption process. Without oxygen, active sites on the photocatalyst’s surface may become more readily occupied as MO concentration increases, inducing more robust interactions and an increased adsorption rate constant. Such a rise may also be ascribed to the reduced competition for adsorption sites when oxygen molecules are absent, allowing MO molecules to adsorb more effectively at higher concentrations. The findings imply that oxygen plays a crucial role in moderating adsorption kinetics, with its absence leading to a concentration-dependent increase in the rate constant. This observation highlights the significant impact of oxygen on the efficiency and behavior of the adsorption process.

The initial adsorption rate h of the PSO model consistently increases under both conditions, with a more significant rise in O_2_-depleted solutions as MO concentration increases. This indicates a higher adsorption rate at elevated concentrations, as more MO molecules reach the HFs surface quickly owing to a higher magnitude of force involved in mass transfer.

Finally, the Elovich model fits well with the data, showing R² values exceeding 0.993, which suggests some degree of surface heterogeneity in the adsorption process. The Bangham, and Weber–Morris models also emphasize that pore and intraparticle diffusion are significant, particularly at higher MO concentrations. A higher k_id_ value at increased initial concentrations indicates that intraparticle diffusion occurs more rapidly, when the MO initial concentration is higher. These findings indicate that when the concentration gradient increases, the driving force for diffusion also increases, resulting in faster adsorption within the pores of the HFs. This behavior is typical because higher concentrations provide more molecules to interact with the available adsorption sites, thereby accelerating the diffusion process.

### 3.4. Photocatalytic Evaluation and Reaction Kinetics of Cu-CuO/TiO_2_ Nanocomposite—First Experimental Cycles

The UV MO photolysis was first carried out under UV-A illumination without a photocatalyst. After 1.5 h of UV exposure, the dye exhibited minimal degradation. Before each photocatalytic experiment, the aqueous MO solution was sparged with O_2_ or He. Figure 5 presents the overall kinetics of MO degradation in O_2_-saturated solutions at the tested concentrations, along with a representative spectrum plot of the attenuation of the main (464 nm) and secondary (271 nm) MO absorbance peaks at a concentration of 10 mg/L during the photocatalytic process using the Cu-CuO/TiO_2_ nanocomposite. The steady attenuation of the MO absorbance peaks at λ = 464 and 271 nm, shown in the inset of Figure 5a, indicates a very fast degradation of the azo dye. Significant reduction of the secondary peak occurs after 90 min of UV illumination, suggesting the breakdown of the aromatic ring. Additionally, there is a noticeable blue shift of the main peak (464 nm) throughout the photodegradation process, indicating that following the reductive cleavage of the azo group via the process suggested by the zero-valent copper, specific chemical changes may take place in the outcome molecules [98].

Both the main (Figure 5a) and secondary (Figure 5b) absorbance peaks demonstrate a decrease in concentration over time, indicating effective MO photocatalytic degradation. As the concentration gains higher values, from 6.3 to 24 mg/L, the degradation rate decreases slightly, but approximately 65% degradation is still achieved for the main peak. Only at the highest initial concentration of 24 mg/L is a slower reduction in C/C_0_ values observed, resulting in 45% degradation. Similarly, the secondary peak initially shows 50% degradation at the lowest concentration, remains stable at about 45% for concentrations ranging from 10 to 18 mg/L, and decreases to 35% at the highest concentration of 24 mg/L. This occurs because higher MO concentrations result in increased competition for the available active sites on the photocatalyst’s surface, leading to a slight decrease in the degradation rate. At lower concentrations, there are sufficient active sites available to accommodate the MO molecules, allowing for efficient photocatalytic degradation. As the concentration increases, the active sites become saturated, reducing the overall efficiency of the degradation process. Overall, the results suggest that the Cu-CuO/TiO_2_ nanocomposite is highly effective at degrading MO in O_2_-saturated environments, even at very high concentrations.

Moreover, Figure 6 illustrates the kinetics of MO degradation in O_2_-depleted solutions using Cu-decorated HFs at various concentrations. Both the main (Figure 6a) and secondary (Figure 6b) absorbance peaks decrease over time, indicating effective photocatalytic degradation of MO under these conditions. The degradation rate is more pronounced at lower concentrations (6.3 and 10 mg/L), with a noticeable decrease in C/C_0_ values over 180 min. The main peak shows 73.5% degradation at 6.3 mg/L concentration and decreases to 40% at the highest concentration (18 mg/L). Similarly, the secondary peak presents a reduction from 62% to about 30% degradation. The inset plot at a concentration of 10 mg/L reports the decrease in both the primary and secondary absorbance peaks during the photocatalytic process, therefore validating the observed degradation pattern.

A comparative analysis of the kinetics of MO degradation in O_2_-saturated and O_2_-depleted solutions highlights differences in the efficiency and behavior of the photocatalytic process using Cu-CuO/TiO_2_ nanocomposite. In O_2_-saturated solutions, both the main and secondary absorbance peaks demonstrate a significant reduction over time, with a noticeable degradation rate even at higher concentrations. This indicates that the presence of oxygen enhances the photocatalytic activity, likely due to the generation of reactive oxygen species, which facilitate the breakdown of MO molecules. This implies that oxygen availability is essential for maintaining high degradation efficiency, as it helps in the continuous regeneration of active sites on the catalyst surface. Another possible explanation could be that dissolved oxygen acts as an electron scavenger, reducing recombination by trapping electrons from the conduction band of the photocatalyst [43]. When the amount of dissolved oxygen is high relative to the electrons generated by photocatalytic activity on the surface, the rate of electron transfer from the catalyst surface to dissolved oxygen will increase significantly. The effect of charge transfer on photocatalysis [99,100] is very important and explains the better photocatalytic efficiency under these conditions.

In contrast, the degradation kinetics in O_2_-depleted solutions show a more pronounced decrease in the degradation rate as the MO concentration increases. While effective photocatalytic degradation is still observed, particularly at lower concentrations of 6.3 mg/L and 10 mg/L, the reduction in absorbance peaks is less significant at higher concentrations. This reduction indicates that, without the presence of oxygen, the photocatalytic activity relies more on the inherent properties of the Cu-CuO/TiO_2_ nanocomposite and the availability of other reactive species. These properties of nanocomposite stem from copper and/or copper oxide nanoparticles, which act as electron sinks (in the form of metallic copper) or by forming p–n CuO/TiO_2_ heterojunctions. This process enhances the separation of electron-hole pairs, as discussed in our previous work [62]. This suggests that the absence of oxygen constrains the production of reactive species and the regeneration of active sites, leading to lower overall degradation efficiency, particularly at higher pollutant concentrations.

While the main focus of this study was the adsorption kinetics and photocatalytic degradation of MO using the developed copper-decorated photocatalyst in the form of HFs, its performance was compared with other copper-titania nanocomposite or doped powders from the literature, as shown in Table 2. Under nearly identical reaction conditions, the photocatalytic efficiency of HFs (with low light intensity) is comparable to those reported in the literature. It is important to note that light intensity is a critical factor, as it significantly influences photocatalytic degradation by determining the number of generated electron-hole pairs [62]. On the other hand, there is always a tradeoff between the high photocatalytic efficiency of slurry systems and the challenges associated with their efficient recovery from solution and effective irradiation. Additionally, most slurry systems exhibit minimal adsorption. In contrast, our developed material demonstrates enhanced adsorption capacity, with its overall removal efficiency clearly approaching that of slurry systems.

Additionally, it has been frequently noted that the photocatalytic degradation rate of various pollutants follows the Langmuir–Hinshelwood (L–H) model [57,101]. For solutions with millimolar concentrations, pseudo-first order kinetics were presumed to compute the related degradation rate constant k_app_ (min^–1^). This method enables the estimation of the degradation speed under specific conditions. Then, for the calculation of the k_app_, the subsequent equations are used:(11)$$\ln\left(\frac{C_{0}}{C}\right)=k_{r}\times K_{LH}\times t=k_{app}\times t$$
where C_0_ is the initial concentration (mg/L), C is the concentration (mg/L) at time t (min), k_r_ is the reaction rate constant (mg/L·min), and K_LH_ is the adsorption constant of the reactant (L/mg). Thus, the initial degradation rate can be determined using the following equation:(12)$$r^{\prime }=k_{app}\times C_{0}$$

Finally, the linear form of the L–H model, represented by the equation below, shows the relationship between the 1/r′ values and the corresponding 1/C_0_ values:(13)$$\frac{1}{r^{\prime }}=\frac{1}{k_{r}}+\frac{1}{k_{r}\times K_{LH}}\times \frac{1}{C_{0}}$$The rate constant is originally determined by plotting the ln(C_0_/C) versus time for the tested different initial MO concentrations. The slope of the linear regression corresponds to k_app_. These results, along with the initial degradation rates, are listed in Table 3.

Table 3 provides insights into the MO degradation kinetics using Cu-CuO/TiO_2_ nanocomposite under both O_2_-saturated and O_2_-depleted conditions across various initial concentrations. In O_2_-saturated conditions, the k_app_ is higher or comparable to that in O_2_-depleted conditions, except at the lowest concentration of 6.3 mg/L, indicating that the presence of oxygen enhances the photocatalytic activity at some extent. There is a peak at 12 mg/L, followed by a decrease, implying that while Cu-CuO/TiO_2_ nanocomposite maintains catalytic efficiency at lower concentrations, its performance diminishes as MO concentration increases due to possible saturation of active sites or limitations in oxygen availability. Conversely, at higher concentrations, the comparison of MO degradation k_app_ values between the O_2_-depleted and O_2_-saturated solutions reveals they are almost identical. Although the Cu-CuO/TiO_2_ nanocomposite presented reduced adsorption capacity towards MO in the O_2_-saturated solution, and despite the fact that dissolved oxygen can act as a scavenger for photogenerated electrons (generating ˙O_2_^–^, instead of the more reactive ˙OH radicals), the photocatalytic efficiency of the nanocomposite remained unaltered. Thus, it can be inferred that the reductive cleavage of the azo group is another mechanism accountable for the enhanced capacity of MO degradation [102].

Moreover, Figure 7 illustrates the plots of the initial degradation rate r’ against the initial MO concentration in both O_2_-saturated and O_2_-depleted solutions, revealing a non-linear relationship. In the O_2_-saturated solution, the initial degradation rate generally remains stable with slight fluctuation until 12 mg/L. Beyond this point, the rate nearly doubles and stabilizes at very high concentrations after the 15 mg/L, indicating optimal conditions for degradation and possible saturation of active sites on the catalyst. This observation implies that the presence of dissolved oxygen enhances the degradation process, likely by providing sufficient electron scavengers to facilitate the reaction. Conversely, in the O_2_-depleted solutions, the initial degradation rate is consistently lower than in the O_2_-saturated solutions, except at a concentration of 6.3 mg/L, and follows a different trend. The rate initially decreases slightly as the concentration increases, suggesting that the lack of oxygen significantly limits the degradation process by reducing the efficiency of electron transfer and scavenging. However, after reaching a minimum point, the rate begins to rise by up to 35% compared to the lowest concentration, indicating that at higher concentrations, other mechanisms might contribute to degradation (e.g., reductive cleavage of the azo group), or that the system approaches a different kinetic regime. Overall, these observations highlight the critical role of dissolved oxygen in influencing the photocatalytic degradation rates and the complex interplay between concentration and reaction kinetics.

Finally, the calculated values for k_ro_ and K_LHo_ values from the slope and intercept of the linear regression in O_2_-saturated solutions were 0.128 mg/L/min and 0.072 L/mg, respectively. According to the L–H model, the identical values of K_LH_ and the Langmuir adsorption constant (b = 0.065 L/mg) confirm the dye’s strong affinity to the catalytic surface, suggesting surface-bound reactions predominate. Again, under O_2_-depleted conditions, the k_ri_ and K_LHi_ values were determined to be 0.042 mg/L/min and 0.334 L/mg, respectively. The differences in K_LH_ and b (0.074 L/mg) suggest that the reaction takes place not only on the surface, but also within the bulk solution that is confined within the pore structure of the Cu-decorated HFs. This implies that the porous structure of the catalyst facilitates additional reaction pathways, contributing to MO degradation even when oxygen is limited.

### 3.5. Half-Time of Reaction

To determine the rate of reaction for first-order kinetics, one of the most practical indicators is the calculation of the half-life time of reaction [70]. The half-life time (denoted as t_1/2_) is the time required for the concentration (C) to reach half of its initial value (C_0_). It can be determined using the following equation [70,101]:(14)$$t_{1/2}=0.5\times \frac{C_{0}}{k_{r}}+\frac{\ln2}{k_{r}\times K_{LH}}$$Moreover, for reactions displaying pseudo-first order kinetics, it is possible to determine the half-life time by employing the following equation:(15)$$t_{1/2}^{\prime }=\frac{\ln2}{k_{app}}$$

All calculated values of t_1/2_ and $t_{1/2}^{\prime }$ are listed in Table 4. As the initial MO concentration increases, the half-life t_1/2_ in O_2_-saturated solutions also increases, indicating slower reaction kinetics, particularly at higher concentrations. In addition, there is generally a consistent difference between t_1/2_ and $t_{1/2}^{\prime }$, which became more pronounced as the initial MO concentration increased. This trend was more gradual in O_2_-saturated solutions and can be attributed to the formation of intermediates that may compete for adsorption on the catalyst, thereby slowing down the reaction kinetics. The data also highlight that the half-life in O_2_-depleted conditions remains more consistent, demonstrating that the absence of oxygen allows for more stable reaction kinetics. Consequently, as a general observation from the kinetic model analysis, it is essential to take into account the competitive influence of intermediate byproducts during the photocatalytic process.

### 3.6. Regeneration and Reusability of Cu-CuO/TiO_2_ Nanocomposite

As aforementioned in Section 2.3, a series of five consecutive runs (regeneration/recycling) was conducted to assess the photocatalytic performance and stability of the developed Cu-CuO/TiO_2_ nanocomposite under UV illumination. At the end of each experimental cycle, the used sample was easily retrieved from the spent MO solution, rinsed twice with ultrapure water, dried, and then reused with a fresh amount of MO solution from the stock for the subsequent experimental cycle. Figure 8a and Figure 8b present the comparison of the MO adsorption capacity and rejection efficiency of a newly prepared sample with a sample subjected to a second experimental cycle in both O_2_-saturated and O_2_-depleted solutions, respectively. In both O_2_-saturated and O_2_-depleted solutions, the regenerated Cu-CuO/TiO_2_ nanocomposite exhibits a slightly lower MO adsorption capacity q_e_ and efficiency R compared to the fresh Cu-CuO/TiO_2_ nanocomposite, as indicated by the declining ratios in the plots, particularly at higher concentrations. This decrease is more significant in O_2_-saturated solutions than in O_2_-depleted ones.

In O_2_-saturated solutions, there appears to be a discernible concentration threshold (15 mg/L of MO) beyond which the rejection performance during the second photocatalytic test slightly decreases. Up to this 15-mg/L concentration, the nanocomposite preserves 93% of its adsorption capacity and 90% of its rejection efficiency. However, at the highest concentration studied, these values drop to approximately 78% and 74% of the initial performance, respectively. In addition, it can be noted that the concentration threshold for the degradation of photocatalytic and adsorption performances coincides, underscoring the critical contribution of adsorption in the photocatalytic process. Beyond this threshold, some active adsorption sites on the Cu-CuO/TiO_2_ nanocomposite become occupied (saturation), and simple washing is insufficient to regenerate them effectively at higher concentrations, resulting in a slight degradation of adsorption and photocatalytic performance. Thus, this observed decrease is likely due to the competitive adsorption between oxygen molecules and MO molecules on the active sites of the catalyst, or the oxidative reactions during the regeneration process that may lead to the degradation or deactivation of these active sites on the nanocomposite. This competition reduces the overall adsorption capacity and efficiency.

In the case of O_2_-depleted solutions, the data from Figure 8b indicate the same concentration threshold (15 mg/L), over which photocatalytic and adsorption performance experience a decline. Similarly, the nanocomposite maintains 84% of its adsorption capacity and 89% of its rejection up to this concentration. Though, at the highest concentration studied, it retains approximately 75% and 83% of the initial performance, respectively. In O_2_-depleted solutions, the Cu-CuO/TiO_2_ nanocomposite has more available active adsorption sites due to reduced competition, allowing MO molecules to adsorb more effectively and preserve its initial performance. This contrasts with O_2_-saturated ones, where dissolved oxygen acts as a scavenger and occupies sites within the residual carbon nest, reducing the accessible sites for MO adsorption. This suggests that the regeneration process impacts the adsorptive properties of the nanocomposite, leading to slightly reduced effectiveness over multiple cycles. Additionally, the reduction in both adsorption capacity and efficiency is more pronounced at higher MO concentrations, indicating that the regeneration process may not fully restore the adsorptive properties of the Cu-CuO/TiO_2_ nanocomposite under higher pollutant loads. The same observations apply to the secondary peak (271 nm, aromatic ring) as well.

Figure 9 illustrates the findings of the recycling analysis, showing the reusability of the Cu-CuO/TiO_2_ nanocomposite as photocatalyst over five successive cycles for various initial MO concentrations in both O_2_-saturated and O_2_-depleted solutions. Under O_2_-saturated conditions (Figure 9a), the photocatalytic performance remains relatively stable across cycles, particularly at lower MO concentrations (from 6.3 up to 12 mg/L), where degradation efficiency remains high. The nanocomposite maintains over 92% performance at a concentration of 15 mg/L and 95% at lower concentrations. As MO concentration increases, there is a slight decline in photocatalytic activity, suggesting that higher concentrations may cause the saturation or occlusion of active areas on the surface of the catalyst, reducing its effectiveness over multiple uses. However, it still manages to maintain 60% of the initial performance at the highest concentration (24 mg/L).

In contrast, under O_2_-depleted conditions (Figure 9b), there is a noticeable reduction in degradation efficiency with successive cycles, especially at higher MO concentrations. Up to 10 mg/L, the nanocoposite retains 95% of their initial efficiency, but beyond 15 mg/L, this drops to about 87% and eventually reaches 72% of the initial performance. This decline could be due to the reduced availability of oxygen, which is crucial for regenerating reactive sites on the catalyst surface. At lower concentrations, the catalyst exhibits better stability and performance across cycles, similar to the O_2_-saturated conditions. Overall, these results indicate that while the Cu-CuO/TiO_2_ nanocomposite is effective photocatalyst with good reusability, its performance is notably enhanced by the presence of oxygen at higher concentrations, which helps maintain photocatalytic efficiency over repeated cycles. Optimizing oxygen levels and managing MO concentrations could further improve the durability and efficacy of this developed photocatalyst in practical applications.

## 4. Conclusions

This work presents a thorough investigation of the adsorption kinetics and photocatalytic MO degradation using Cu-CuO/TiO_2_ nanocomposite in the form of HF as an advanced photocatalyst. The results demonstrate that MO adsorption capacity is highly dependent on the initial MO concentration, with higher concentrations leading to increased adsorption but slower equilibrium. In both O_2_-saturated and O_2_-depleted environments, the adsorption process appears to be primarily driven by physical mechanisms, which are further enhanced by the structural properties of the Cu-decorated HFs. The presence of residual carbon and copper and/or copper oxide nanoparticles improves their pore structure and surface characteristics, facilitating more effective adsorption through π-π interactions and electron donor–acceptor mechanisms.

The kinetic analysis reveals that the experimental data are most accurately described by the PSO model, although the PFO model more accurately predicts the adsorption capacity, suggesting that the adsorption process is likely governed by a physical adsorption mechanism rather than a chemical one. The study also underscores the significant influence of pore and intraparticle diffusion in the adsorption process, especially at higher MO concentrations. As the concentration gradient increases, the driving force for diffusion also rises, resulting in faster adsorption within the HFs pores. The Langmuir–Hinshelwood model applied to the reaction kinetics confirms that the photocatalytic performance of the Cu-CuO/TiO_2_ nanocomposite is impacted by both MO concentration and oxygen availability, with higher k_app_ values observed under O_2_-saturated conditions. Furthermore, the presence of intermediate byproducts during the photocatalytic process also plays an important role in the overall degradation efficiency.

The photocatalytic evaluation of the Cu-CuO/TiO_2_ nanocomposite reveals its high efficiency in degrading MO, especially in O_2_-saturated environments, where oxygen enhances the degradation rate via the production of reactive oxygen species. The degradation remains consistent across various concentrations, with approximately 65% degradation achieved for most concentrations and a slight decrease observed only at the highest concentration of 24 mg/L. In O_2_-depleted solutions, the initial degradation rates are higher at lower concentrations (73.5% and 62% for the main and secondary peaks, respectively) but decrease to around 40% and 30% at 18 mg/L. This suggests that the inherent properties of the Cu-decorated HFs, such as the formation of p–n CuO/TiO_2_ heterojunctions, contribute to the degradation process, but the absence of oxygen limits their performance, particularly in more concentrated solutions.

The regeneration and reusability assessment of the Cu-CuO/TiO_2_ nanocomposite indicates its potential for repeated use, with only a minor performance decline observed over multiple cycles up to 15 mg/L. The photocatalytic activity remains relatively stable, particularly at lower MO concentrations. However, the efficiency decreases gradually at higher concentrations, possibly due to the saturation or deactivation of active sites. Despite this, the Cu-CuO/TiO_2_ nanocomposite still retains 60% of its initial performance at the highest concentration of 24 mg/L in O_2_-saturated solutions and 72% in O_2_-depleted solutions. The regeneration process impacts the adsorptive properties of the nanocomposite, with more pronounced effects in O_2_-depleted conditions due to limited oxygen availability for site regeneration.

Overall, this study represents a significant advancement in the development of photocatalytic systems for pollutant removal. The proficiently engineered HF photocatalyst of Cu-CuO/TiO_2_ nanocomposite offers a promising approach for addressing environmental challenges associated with organic contaminants through its combined adsorption and photocatalysis capabilities. This study paves the way for the practical implementation of advanced photocatalysts in real-world environmental applications, including wastewater treatment and remediation, through their potential use in continuous flow industrial processes.

## Figures and Tables

**Figure 1 materials-17-04663-f001:**
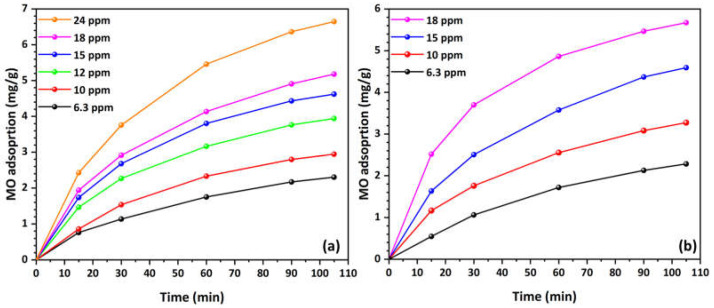
MO adsorption kinetics curves at various concentrations in (**a**) O_2_-saturated and (**b**) O_2_-depleted solutions (natural pH, 25 °C).

**Figure 2 materials-17-04663-f002:**
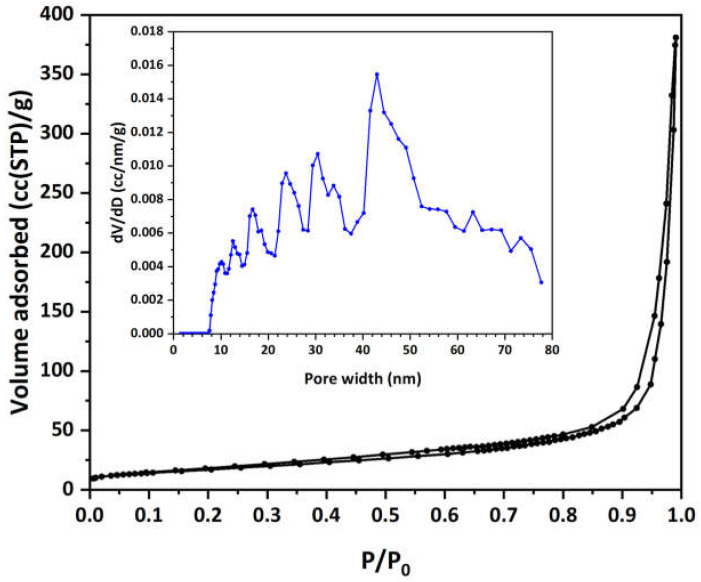
N_2_ adsorption-desorption isotherm (77 K) of the prepared Cu-CuO/TiO_2_ nanocomposite material. In the inset, the pore size distribution (PSD) curve obtained from the desorption branch of LN_2_ isotherm using the NLDFT method is displayed.

**Figure 3 materials-17-04663-f003:**
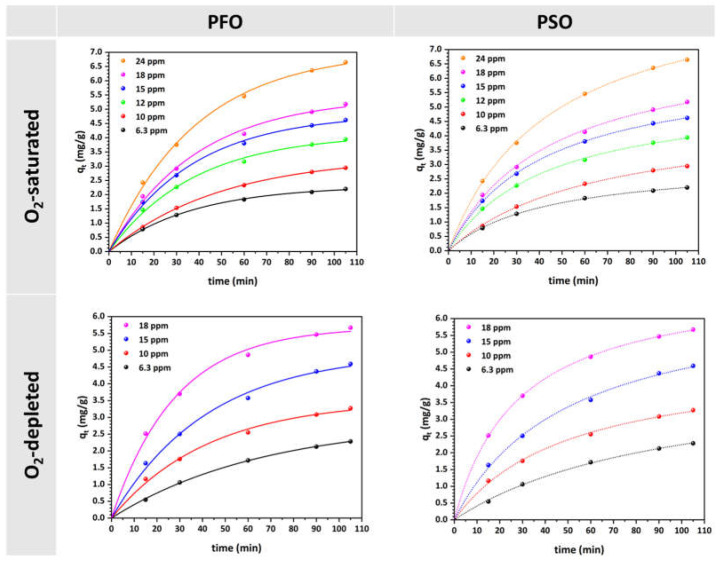
Pseudo-first order (PFO) and pseudo-second order (PSO) model plots for MO uptake on the Cu-CuO/TiO_2_ nanocomposite at different initial concentrations in O_2_-saturated and O_2_-depleted solutions (natural pH, 25 °C).

**Figure 4 materials-17-04663-f004:**
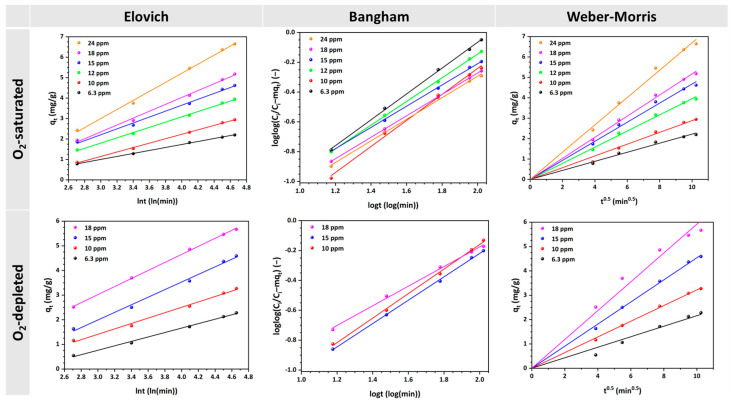
Elovich, Bangham, and Weber–Morris model plots for MO uptake on the Cu-CuO/TiO_2_ nanocomposite at different initial concentrations in O_2_-saturated and O_2_-depleted solutions (natural pH, 25 °C).

**Figure 5 materials-17-04663-f005:**
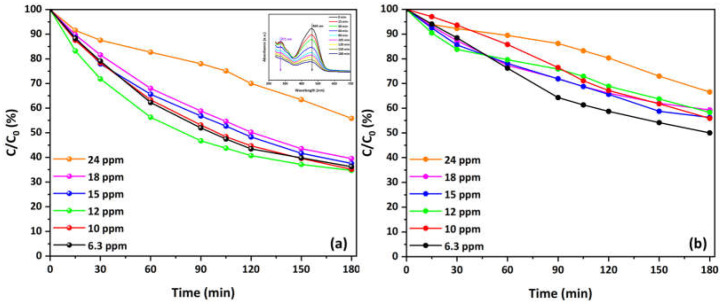
Overall kinetics of MO degradation in O_2_-saturated solutions at various concentrations using the Cu-CuO/TiO_2_ nanocomposite: (**a**) main and (**b**) secondary peak. The inset presents a representative spectrum plot of the reduction in the main and secondary MO absorbance peaks during the photocatalytic process at C = 10 mg/L (natural pH, 25 °C).

**Figure 6 materials-17-04663-f006:**
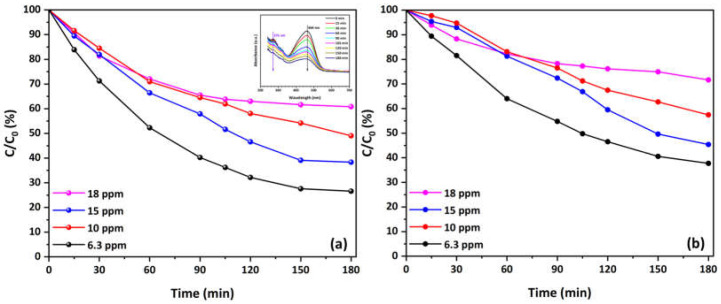
Overall kinetics of MO degradation in O_2_-depleted solutions at various concentrations using the Cu-CuO/TiO_2_ nanocomposite: (**a**) main and (**b**) secondary peak. The inset presents a representative spectrum plot of the reduction in the main and secondary MO absorbance peaks during the photocatalytic process at C = 10 mg/L (natural pH, 25 °C).

**Figure 7 materials-17-04663-f007:**
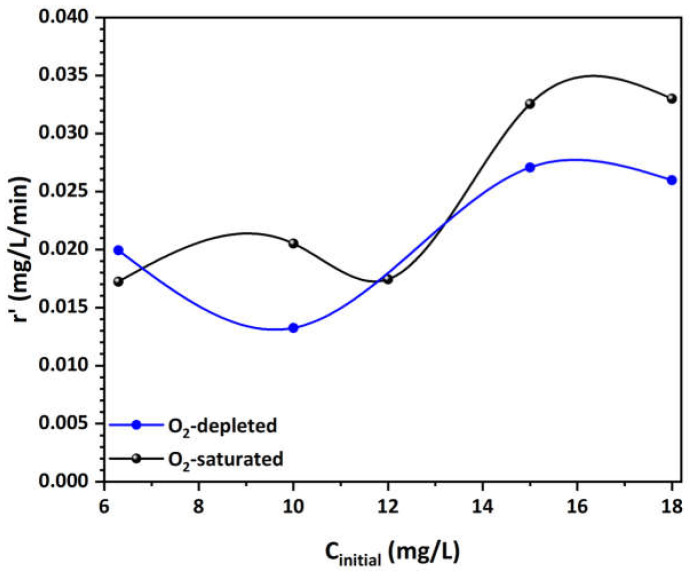
Effect of initial MO concentration on the initial degradation rate in both O_2_-saturated and O_2_-depleted solutions.

**Figure 8 materials-17-04663-f008:**
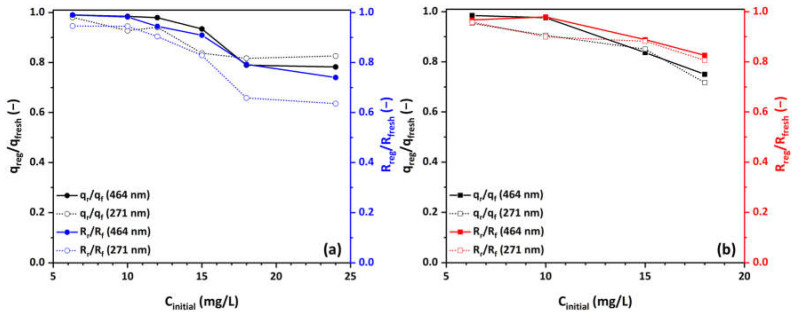
Comparison of the ratio of MO adsorption capacity q_e_ and efficiency R between regenerated and fresh Cu-CuO/TiO_2_ nanocomposite at various MO concentrations in (**a**) O_2_-saturated and (**b**) O_2_-depleted solutions.

**Figure 9 materials-17-04663-f009:**
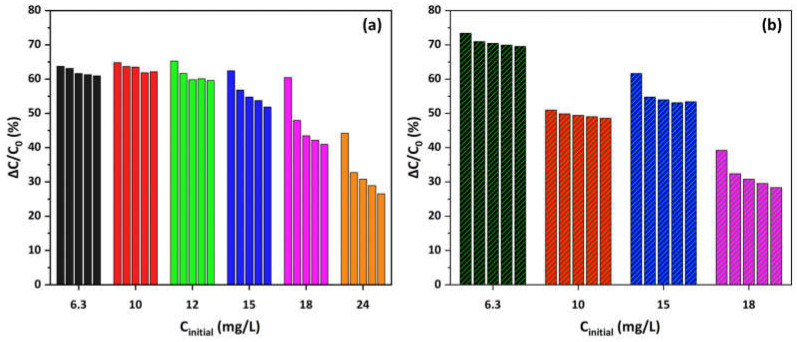
MO rejection (%) due to the photocatalytic degradation process by the Cu-CuO/TiO_2_ nanocomposite under UV-A illumination after five successive cycles at various concentrations in (**a**) O_2_-saturated and (**b**) O_2_-depleted solutions (natural pH, 25 °C).

**Table 1 materials-17-04663-t001:** Kinetic parameter data obtained from PFO, PSO, Elovich, Bangham, and Weber–Morris models for MO adsorption on the Cu-CuO/TiO_2_ nanocomposite.

Kinetic Model	Parameters	Solution	Concentration (mg/L)					
			6.3	10	12	15	18	24
**Experimental**	**q_e_ (mg/g)**	**O_2_-saturated**	2.30	2.94	3.94	4.62	5.18	6.65
		**O_2_-depleted**	2.28	3.27	-	4.59	5.67	-
**PFO**	**q_e_ (mg/g)**	**O_2_-saturated**	2.31	3.34	4.13	4.85	5.44	7.05
	**k_1_ (min^–1^)**		0.0270	0.0202	0.0266	0.0273	0.0260	0.0259
	**R^2^ (−)**		0.996	0.9999	0.9972	0.9984	0.9964	0.9988
	**q_e_ (mg/g)**	**O_2_-depleted**	2.88	3.51	-	4.93	5.69	-
	**k_1_ (min^–1^)**		0.0150	0.0235	-	0.0237	0.0357	-
	**R^2^ (−)**		0.9996	0.9963	-	0.9963	0.9972	-
**PSO**	**q_e_ (mg/g)**	**O_2_-saturated**	3.10	4.79	5.53	6.46	7.30	9.51
	**k_2_ (g/mg·min)**		0.00756	0.00324	0.00420	0.00371	0.00311	0.00234
	**h (mg/g·min)**		0.07	0.07	0.13	0.15	0.17	0.21
	**R^2^ (−)**		0.9998	0.9995	0.9995	0.9999	0.9992	0.9998
	**q_e_ (mg/g)**	**O_2_-depleted**	4.40	4.81	-	6.75	7.17	-
	**k_2_ (g/mg·min)**		0.00236	0.00410	-	0.00295	0.00497	-
	**h (mg/g·min)**		0.05	0.09	-	0.13	0.26	-
	**R^2^ (−)**		0.9992	0.9988	-	0.9988	0.9999	-
**Elovich**	**α (mg/g·min)**	**O_2_-saturated**	0.14	0.16	0.26	0.31	0.34	0.43
	**β (g/mg)**		1.37	0.92	0.78	0.66	0.59	0.45
	**R^2^ (−)**		0.9992	0.9989	0.9984	0.9988	0.9966	0.9980
	**α (mg/g·min)**	**O_2_-depleted**	0.10	0.20	-	0.28	0.52	-
	**β (g/mg)**		1.11	0.91	-	0.65	0.61	-
	**R^2^ (−)**		0.9962	0.9936	-	0.9941	0.9990	-
**Bangham**	**α (−)**	**O_2_-saturated**	0.868	0.874	0.795	0.754	0.719	0.725
	**k_0_ (mL/g/L)**		0.437	0.284	0.508	0.540	0.538	0.500
	**R^2^ (−)**		0.9991	0.9968	0.9996	0.9992	1.0000	0.9980
	**α (−)**	**O_2_-depleted**	1.145	0.821	-	0.782	0.657	-
	**k_0_ (mL/g/L)**		0.131	0.437	-	0.456	0.896	-
	**R^2^ (−)**		0.9988	0.9984	-	0.9993	0.9973	-
**Weber–Morris**	**k_id_ (mg·g^−1^·min^−0.5^)**	**O_2_-saturated**	0.2221	0.2883	0.3956	0.4672	0.5172	0.6692
	**R^2^ (−)**		0.9981	0.9968	0.9992	0.9988	0.9995	0.9988
	**k_id_ (mg·g^−1^·min^−0.5^)**	**O_2_-depleted**	0.2160	0.3222	-	0.4541	0.5922	-
	**R^2^ (−)**		0.9921	0.9996	-	0.9996	0.9952	-

**Table 2 materials-17-04663-t002:** Comparison of the photocatalytic performance of the Cu-CuO/TiO_2_ nanocomposite with literature data.

Type	Catalyst Amount (g/L)	Light Intensity (mW/cm^2^)	Results	Reference
doped	1	n/a * (vis)	R ≈ 45% (10 mg/L, 3 h)	[36]
doped	n/a	n/a	R = 73% (50 mg/L, 0.5 h)	[37]
doped	n/a	n/a	R = 82% (9.8 mg/L, 2 h)	[38]
nanocomposite	1	n/a	R = 39.1% (40.9 mg/L, 2 h)	[39]
nanocomposite	1	n/a	R = 80% (20 mg/L, 3 h)	[40]
nanocomposite	2.5	0.5	R = 65.3% (O_2_, 12 mg/L, 3 h)	This work
			R = 73.5% (Inert, 6.3 mg/L, 3 h)	

* n/a: not available.

**Table 3 materials-17-04663-t003:** Degradation rate constant and initial degradation rate values for the Cu-CuO/TiO_2_ nanocomposite used in MO degradation at different concentrations under both O_2_-saturated and O_2_-depleted conditions.

C_initial_ (mg/L)	k_app__o × 10^2^ (min^–1^)	r′_o (mg/L/min)	k_app__i × 10^2^ (min^–1^)	r′_i (mg/L/min)
6.3	0.79	0.017	1.09	0.020
10	0.78	0.021	0.57	0.013
12	0.81	0.017	-	-
15	0.73	0.033	0.68	0.027
18	0.65	0.033	0.68	0.026
24	0.38	0.028	-	-

**Table 4 materials-17-04663-t004:** Calculated half-life reaction times for the Cu-CuO/TiO_2_ nanocomposite at the examined initial MO concentrations in both O_2_-saturated and O_2_-depleted solutions.

C_initial_ (mg/L)	t_1/2_o_ (min)	t_1/2_i_ (min)	t′_1/2_o_ (min)	t′_1/2_i_ (min)	Δt_1/2_o_ (min)	Δt_1/2_i_ (min)
6.3	83.29	70.93	87.67	63.36	4.38	n/a
10	85.10	76.87	89.31	121.58	4.22	44.71
12	83.17	-	85.42	-	2.25	-
15	92.13	96.46	94.72	101.74	2.59	5.27
18	94.48	94.62	106.08	101.91	11.60	7.29
24	103.56	-	181.47	-	77.91	-

## Data Availability

Data available on request from the authors.

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
