# Peer review of "Investigation of MO Adsorption Kinetics and Photocatalytic Degradation Utilizing Hollow Fibers of Cu-CuO/TiO_2_ Nanocomposite"

_materials, 2024, doi:10.3390/ma17184663_

Round 1

Reviewer 1 Report

Comments and Suggestions for Authors

The article "Investigation of MO adsorption kinetics and photocatalytic degradation using a proficiently engineered HF photocatalyst "

In this article, the Authors explore the kinetics of methyl orange (MO) adsorption and its photocatalytic degradation using an advanced titania-based and copper-decorated photocatalyst in the form of composite hollow fibers (HFs). The photocatalytic decomposition experiments are interesting, but it is important to note that many materials for photocatalytic decomposition have already been developed, and the novel materials are capable of decomposing organic molecules under visible light illumination. The introduction is too broad and not specific for such a study. The materials used in this study are not characterized, and reference 43 does not cover it sufficiently to explain the photocatalysis results.

1)     Hollow fibers (HFs) are an extremely broad class of materials. This study specifically investigated titania-based hollow structures decorated with copper. The study should explicitly focus on this material instead of the whole HF class of materials.

2)     This study lacks SEM, XRD, and Raman spectroscopy characterization. Typically, the black titania forms under the conditions described in the MM section. Moreover, this study uses copper as a decoration, which can act as a dopant. Other analysis would be beneficial, too. The review on black titania synthesis and characterization might help the Authors to analyze the material characterization data: Crystals 2024, 14(7), 647; https://doi.org/10.3390/cryst14070647 and DOI: 10.1039/D1NA00477H (Review Article) Nanoscale Adv., 2021, 3, 5487-5524

3)     It should be proved what structures of copper are forming. Is it copper oxide on top of titania, or titania doped by copper?

4)     The introduction should be rewritten and focused on titania and copper structures (composites, if it is a composite in this study).

5)     Photocatalysis results should be compared to the literature using copper-modified titania.

6)     What is the bandgap of the material? Might it work under visible light after all modifications and modifications with copper?

7)     The MO decomposes under UV light. The control measurements is needed to prove the effect of materials used in this study.

This study is highly interesting; however, more analysis and explanations are needed prior to suggesting for publication.

Reviewer 2 Report

Comments and Suggestions for Authors

This study represents a significant advancement in the development of photocatalytic systems for pollutant removal. The proficiently engineered HF photocatalyst offers a promising approach for addressing environmental challenges associated with organic contaminants through its combined adsorption and photocatalysis capabilities. The experiment is systematically performed, and the results are reasonably discussed. So, the work can be suggested for publication. Some minor revisions are suggested before publication.

1.     The length of paper can be shortened for quick reading. The method for building isotherm model can be moved to supporting information.

2.     The SEM or TEM of HF, Cu modified HF are suggested to be provided.

3.     Do the authors characterize the used Cu/HF, do you confirm the formation of CuO by characterizing?

4.     The effect of charge transfer on photocatalysis should be considered and discussed. Some important references on photocatalytic degradation of organics can help discuss the paper, for example:

10.3866/PKU.WHXB202112027

10.3866/PKU.WHXB202209037

Reviewer 3 Report

Comments and Suggestions for Authors

The investigators conducted a comprehensive study on the adsorption kinetics and methyl orange (MO) degradation via photocatalysis using Cu-decorated hollow fibers. This study is crucial for advancing our understanding of the mechanistic conditions during photocatalytic processes. A comparative analysis under O₂-saturated and O₂-depleted conditions highlighted the significance of physical mechanisms within the photocatalytic reaction system. The kinetic analysis, employing various models, revealed the importance of physical adsorption - a factor often overlooked in many studies. The results are scientifically significant, and I recommend this work for publication in Materials. However, to ensure the validity of the experimental results, additional experimental details should be provided, and the following considerations should be addressed.

1.      The presence of trace metals or impurities may have influenced the observed behaviour. To evaluate this, the authors should provide the details regarding the purity of water used in experiments.

2.      The chemical formula of all chemicals used should be included alongside their names in the materials section. Also, those materials where only chemical formula was mentioned, their names should be provided. The % purity of the chemicals should be specified as well.

3.      During the synthesis process, ethanol, glutaraldehyde and HCl were used. The quantities of these chemicals or the ratio of their concentrations should be provided.

4.      Also, the concentration of copper nitrate in the solution should be clearly stated.

5.      Prior to UV radiation, the catalyst containing dye solutions were stirred in the dark. The authors should specify the duration of this process.

6.      To test the recycling process, the authors mentioned that the used samples were retrieved. They should detail the method used for sample retrieval.

7.      The authors discussed the MO adsorption curves. They should provide information on methodology of these experiments.

8.      TiO2 was commercially purchased and used in the slurry to prepare the HF catalysts. What is the size of them?

9.      Did the authors investigate whether TiO2 has the impact on the catalytic process?

10.  Despite multiple exposed phases, the Langmuir model suggested that adsorption occurred at homogeneous sites. Did the authors performed physical characterization to check how the materials were distributed in the catalyst? If they are not homogeneous, isolated sites of specific material might be the only active ones in this reaction. It would be valuable to know material is active and why others did not show any activity.

11.  The BET of the samples was calculated in the previous work. Are the samples used in these reactions from the same batch as those in earlier study? If different, BET may vary between different batch.

12.  The authors suggested that Cu might impact catalysis by acting as an electron sink, possibly through the formation of a p-n CuO/TiO2 junction. This is contradictory to the results from Langmuir model, which indicated homogeneous adsorption sites. The authors should address this discrepancy

13.  The authors noted that O2 saturated conditions are always good, which is expected. They also mentioned that catalysis occurred not only by the adsorption on the catalyst but also in the bulk solution trapped within the catalyst pores. This could occur if there are free radicals present in the solution. However, the models investigated suggest that an additional mechanism may also be happening. Did the authors attempt to test this by using H2O2, where the reaction is mainly driven by radicals?

Comments on the Quality of English Language

It is good. minor typo errors have to be rectified

Round 2

Reviewer 1 Report

Comments and Suggestions for Authors

Dear Authors,

thank you for the comments and improvements. All the remarks from the review report were discussed.

The manuscript can be suggested for publication.